# Infection Prevention and Control Strategies According to the Type of Multidrug-Resistant Bacteria and *Candida auris* in Intensive Care Units: A Pragmatic Resume including Pathogens R_0_ and a Cost-Effectiveness Analysis

**DOI:** 10.3390/antibiotics13080789

**Published:** 2024-08-22

**Authors:** Chiara Fanelli, Laura Pistidda, Pierpaolo Terragni, Daniela Pasero

**Affiliations:** 1Department of Medicine, Surgery and Pharmacy, University of Sassari, 07100 Sassari, Italylpistidda@uniss.it (L.P.); pterragni@uniss.it (P.T.); 2Head of Intensive Care Unit, University Hospital of Sassari, 07100 Sassari, Italy; 3Head of Intensive Care Unit, Civil Hospital of Alghero, 07041 Alghero, Italy

**Keywords:** infection prevention and control, hospital-acquired infections, outbreak, multidrug-resistant bacteria, *Acinetobacter baumanii*, *Candida auris*, VRE, KPC, basic reproduction number, decolonization

## Abstract

Multidrug-resistant organism (MDRO) outbreaks have been steadily increasing in intensive care units (ICUs). Still, healthcare institutions and workers (HCWs) have not reached unanimity on how and when to implement infection prevention and control (IPC) strategies. We aimed to provide a pragmatic physician practice-oriented resume of strategies towards different MDRO outbreaks in ICUs. We performed a narrative review on IPC in ICUs, investigating patient-to-staff ratios; education, isolation, decolonization, screening, and hygiene practices; outbreak reporting; cost-effectiveness; reproduction numbers (R_0_); and future perspectives. The most effective IPC strategy remains unknown. Most studies focus on a specific pathogen or disease, making the clinician lose sight of the big picture. IPC strategies have proven their cost-effectiveness regardless of typology, country, and pathogen. A standardized, universal, pragmatic protocol for HCW education should be elaborated. Likewise, the elaboration of a rapid outbreak recognition tool (i.e., an easy-to-use mathematical model) would improve early diagnosis and prevent spreading. Further studies are needed to express views in favor or against MDRO decolonization. New promising strategies are emerging and need to be tested in the field. The lack of IPC strategy application has made and still makes ICUs major MDRO reservoirs in the community. In a not-too-distant future, genetic engineering and phage therapies could represent a plot twist in MDRO IPC strategies.

## 1. Introduction

Hospital-acquired infections (HAIs) are a major concern for public health and a major issue in ICUs [1]. HAIs are defined as infections acquired after hospitalization that manifest themselves 48 h after admission to the hospital. The most common HAIs include ventilator-associated pneumonia (VAP), central line-associated bloodstream infections (CLABSIs), and catheter-associated urinary tract infections (CAUTIs) [1].

According to the WHO’s Global Report on Infection Prevention and Control (IPC) of 2022, 7% of patients in high-income countries and 15% of patients in low- and middle-income countries (LMICs) acquire at least one HAI during hospitalization [2,3].

These numbers rise dramatically if we take into consideration only adult ICUs: almost one out of three (30%) hospitalized patients develop an HAI; in fact, almost half of all cases (48.7%) of sepsis with organ dysfunction treated in ICUs are hospital-acquired [2,4], resulting in death in 52.3% [4].

In particular, MDROs’ prevalence in ICUs is currently a major concern as they can account for up to 50% of all infections in this setting [5]. In the USA, more than 700,000 HAIs each year are associated with MDR bacteria in ICUs, while in Europe, a growing rate of HAIs caused by *Carbapenemase*-*producing Enterobacteriaceae* (CPE) and *New Delhi Metallo-Betalactamase* (NDM) producers have been observed. However, the greatest threat in ICUs is currently represented by MDR *Acinetobacter baumannii*, *Pseudomonas*, *Enterobacteriaceae*, and *Candida auris*, against which there are few weapons available [6].

These global estimates of HAI frequency are probably downgraded by several factors: the lack of HAI surveillance and outbreak reporting systems, poor adherence to standardized protocols, and poor study quality [2].

The reason for the higher percentages in ICUs is grounded only partially in its intrinsic risk factors for infection acquisition (i.e., the use of invasive devices, high severity of acute illness, predisposing underlying conditions, and being at age extremities [7]); the lack of infection control is the real killer.

According to the WHO, national infection prevention and control (IPC) programs and operational plans are not available in all countries and, when available, they may not be fully implemented. This is the case for Italy and Romania in Europe; Bolivia, Costa Rica, and Honduras in South-Central America; and India, Nepal, Bhutan, Myanmar, Tagikistan, Turkmenistan, Iraq, and Afghanistan in Asia [2]. In Africa, 15 out of 54 states find themselves in this condition [2].

These programs are a key element to put IPC into practice and prevent *multidrug-resistant organisms* (MDROs) from spreading in and outside of the hospital setting.

IPC is based on two different but complementary approaches: targeted and universal approaches. The targeted approach consists of screening and isolation, and the details are usually contained in a bundle.

The aim of this narrative review of the literature is to summarize and display the most successful and pragmatic strategies to achieve infection control in ICU outbreaks.

## 2. Methodology

This review reports the major risk factors for HAI acquisition identified in the ICU setting and proposes strategies in guidelines, the WHO’s recommendations, international institutional statements, and outbreak reports in the past 25 years, although it does not comprehensively report on all of the literature as a systematic review would. The choice of 25 years is dictated by the will to include only the most consolidated strategies and novelties in the field of IPC, including those specifically against MDROs. Every statement and statistic reported in the following section of this paper refers to adult ICU departments, unless otherwise marked.

The search was conducted on the PubMed electronic database and included only peer-reviewed articles. No language restriction was applied. Publications were firstly screened by title, abstract, and year of publishing by CF. Afterwards, CF evaluated the full articles in order to assess the eligibility for inclusion, and they were consequently reviewed by DP, LP, and PT. The quality of the data and accuracy of description of the proposed strategy, together with the novelty, were considered as the most weighted factors in the selection process.

All investigated variables in the context of IPC (patient-to-staff ratios, education, isolation, decolonization, screening and hygiene practices, outbreak reporting, cost-effectiveness, reproduction-number (R_0_), and future perspectives) were selected on the basis of (1) recurrency in IPC guidelines and WHO statements, (2) frequency of discussion and solution proposals in outbreak reports, and (3) real-life, daily faced IPC problematic issues authors experience as an infectologist (CF) and intensivists (DP, LP, and PT).

## 3. Outbreaks Genesis

An HAI outbreak could be defined as an increased number of cases of a certain HAI among patients or healthcare personnel that is superior to the expected number, which is clustered by time and place [8,9,10].

Transmission occurs differently depending on the pathogen, involving environmental, healthcare organization-dependent, laboratory, and host-dependent factors [7].

The routes of transmission for some of the most common pathogens isolated in ICUs [11] are displayed in Table 1. The hematogenous route was not listed as the routinary use of gloves for invasive procedures is commonly adopted and effectively prevents the spread of blood-borne diseases.

Although outbreaks involve a large number of individuals, the risk factors for HAI acquisition should be taken into consideration for both the type of patient and type of infection.

## 4. Risk Factors for Outbreak

Outbreaks generally depart from a non-diagnosed infected or colonized patient for a transmittable disease [7]. Therefore, the first risk factor is represented by the lack of diagnosis.

A rapid outbreak recognition tool (i.e., an easy-to-use mathematical model) should be proposed to improve early diagnosis and prevent spreading. For example, the identification of three cases in 5 days could be an outbreak triggered test, as experimented by Elliot et al. [28].

There are several factors, both generic and pathogen-specific, that interlude the outbreak genesis. All generic risk factors are listed as separate paragraphs in this review, including patients’ colonization (Box 1) [12,27,29] management, and pathogen-specific risk factors are listed as sub-paragraphs or the main focus.

Box 1MDRO colonization.
It is defined by the presence of an MDRO without the evidence of tissue invasion or associated symptoms:○Regular sites: Respiratory secretions (nostrils, pharynx, and endotracheal aspirates), wounds, skin, urine, and the rectum. More than one site could be affected by the same colonization.○Sterile sites (not interpretable as colonization): Blood, liquor, pleural, peritoneal, and synovial liquids.A colonized patient always represents a potential source of transmission.It requires contact and/or respiratory isolation beyond routine IPC procedures.Decolonization is not currently recommended in the ICU setting.


Among the most underrated, artificial fingernails have been associated with HAIs, such as Serratia marcescens bloodstream infections (BSIs) in patients undergoing hemodialysis [30] and ESBL-producing *Klebsiella pneumoniae* and *Pseudomonas aeruginosa* invasive infections in neonatal ICUs [31].

Notably, a body mass index (BMI) ≥30 and an elevated number of hospitalizations have also been associated with a major risk of acquiring MRSA [32,33], CRE [34], and VRE [34] colonization.

Intravenous and inhalation drug use is the important risk factor to community-acquired MRSA colonization; therefore, they have to be screened at admission [32].

## 5. Strategies

IPC strategies are multiple and synergic. All variables that are worth considering for the purpose of a successful infection control process are reported below. They include the patient-to-nurse ratio (PNR), patient-to-intensivist ratio (PIR), healthcare staff education, isolation types, MDRO decolonization, hand hygiene, shoe hygiene, screening, environmental cleaning, antimicrobial stewardship programs, outbreak reporting, special populations, cost-effectiveness and R_0_, new experiment strategies, and future perspectives. The quality of evidence and strength of these practices according to the pathogen are listed in the ESCMID’s guidelines for Infection Control 2014 [12]. To our knowledge, no further updates of these guidelines have been published.

## 6. Nurse-to-Patient Ratio

The connection between nurse- and intensivist-to-patient ratios and infection prevention and control (IPC) is of paramount importance. Hospital IPC is a sanitary problem involving the whole hospital structure [35]. In fact, IPC strategies are not only for first-line healthcare personnel’s utility, but also for healthcare management personnel. Isolation, decolonization, screening and hygiene practices, and outbreak reporting may not be implemented or implemented incorrectly if the healthcare staff is understaffed or not properly acquainted with IPC and healthcare-associated infections (HCAIs) [36,37]. This results in a lack of infection control, beyond a of lack infection prevention, with associated costs in terms of human lives and economic expenditure [38].

There are solid studies in the literature and strong guidelines regarding the patient-to-nurse ratio (PNR) (Table 2). This ratio should be 1:1 or 1:2 according to the kind of ICU. Several international organizations have stated that in the ICU setting, every patient must have immediate access to an ICU specialist nurse, suggesting a PNR of 1:1.

## 7. Physician/Patient Ratio

Currently, there is no clear recommendation on the *patient-to-intensivist ratio* (PIR) by actual guidelines (Table 3). 

Five studies have been published on this topic before Jeremy M. Kahn et al. tried to give an answer to this question in 2023 with a multicenter cohort study on 29 ICUs in 10 hospitals in the United States of America [47]. They failed to find an association between a higher intensivist-to-patient ratio and higher mortality.

Neuraz et al., in 2015, were the first to find an association with the PIR, namely a two-fold increase in shift-specific mortality among French ICU patients cared for by doctors with > 14 vs. < 8 patients [42]. Moreover, Gershengorn HB et al. conducted a similar study in the United Kingdom in 2017, finding a positive association between the PIR (*patient-to-intensivist ratio*) and British ICU patients’ mortality [43]. Five years later, Georgshengorn et al. repeated the study on Australian and New Zealand ICUs, but no association with the PIR was found [45].

Furthermore, studies conducted on this topic in the USA always fail to find an association between the PIR and ICU mortality.

However, as Kerlin MP and Caruso P. stated in their paper and as Kahn et al. pointed out, all five studies that preceded theirs suffered from several methodological limitations [47,49]. For instance, they took into consideration the intensivist-to-patient ratios averaged over the length of the entire ICU stay, overlooking that the ICU census changes day by day and that it could obscure daily variations, which could influence the outcomes. Moreover, they generally extrapolated intensivist-to-patient ratios from ICU census data, neglecting that intensivists may provide care in multiple ICUs within a single day. Additionally, in all of these studies, the majority of intensivists were anesthesiologists/intensivists, but many other specialists, ranging from approximately 10% to 30%, belonged to different medical specialties.

## 8. Education

Beyond the nurse- and physician-to-patient ratios, HCW education on IPC is what affects infectious disease transmission and relative associated mortality the most.

Therefore, not only nurses, but all healthcare personnel (physicians, healthcare workers, medical and nursing students, and cleaning staff) [50] should undergo an ‘IPC course’ as soon as they are hired by the hospital, just before taking an active part in ward activities [50,51]. Furthermore, a ‘refresh IPC course’ should be taken periodically, established by hospital protocols, no less than once per year (or per month, according to local epidemiology). The frequency of the ‘refresh IPC course’ should be rapidly implemented in the case of an outbreak [52].

IPC courses should provide information on pathogens’ transmission, isolation and hand hygiene instructions, and a practical simulation of the procedures. An initial and final practice test should be performed in order to verify the effectiveness of the course and awareness achieved among the healthcare personnel.

The Cochrane Effective Practice and Organisation of Care (EPOC) group elaborated on a seven-item educational model to enhance the uptake of educational contents (Box 2).

Box 2The seven educational strategies elaborated on by the Cochrane Effective Practice and Organisation of Care group [53] to promote the uptake of guidelines.
Printed educational materialsEducational meetingsEducational outreachLocal opinion leadersAudit and feedbackComputerized remindersTailored interventions


Currently, when IPC educational and training programs are present, they differ consistently among WHO countries [54] and are rarely provided by academic institutions, and frequently practicing IPC physicians are not specialized in infectious disease or clinical microbiology [54]. The WHO’s latest guidelines on the core components of IPC programs [50] (Box 3) suggest different and targeted training for each of the identified three categories of HCWs: IPC specialists, HCWs involved in patient care (i.e., nurses and healthcare assistants), and auxiliary personnel (cleaning, administrative, and managerial staff). 

Box 3Core components of IPC (infection prevention and control) programs at the national and acute healthcare facility levels according to the WHO’s guidelines of 2016.
IPC programsIPC guidelines (both at the national level and facility level)IPC education and trainingHealthcare-associated infection (HCAI) surveillanceMultimodal strategies for implementing IPCIPC monitoring, evaluation, and feedbackWorkload, staffing, and bed occupancy (at the facility level)Built environment, materials, and equipment for IPC (at the facility level)


No standardized, universal, pragmatic education protocol has been elaborated so far, so we reported some valuable examples (Table 4).

In addition to implementing proper nurse- and intensivist-to-patient ratios, an educational model on IPC should be adopted by the institution in order to assure the good quality of IPC strategy implementation.

## 9. Isolation

The isolation of the colonized/infected patient is a key strategy for infection control [12]. Without isolation, the other IPC approaches may not be sufficient.

According to the ESCMID’s guidelines of 2014, precautionary isolation for patients recently admitted in the ICU should always be performed in order to avoid the uprise of infection clusters among ICU patients and staff and further hospital clusters [12]. Isolation should be discontinued only after a negative result is obtained from screening procedures (see SCREENING section).

Isolation rooms should preferably be single rooms whenever possible. It is mandatory to provide a single room in the case of neutropenic patients or specific airborne diseases (measles, varicella virus, and tuberculosis) [7].

There are three kinds of isolation: contact isolation, respiratory isolation, and both.

The kind of isolation that should be adopted varies depending on where the pathogen was isolated.

### 9.1. Respiratory Isolation

Respiratory isolation is required every time a respiratory airway sample (rhinopharyngeal swabs, sputum, and bronchial airway liquid fluid or aspirate) has a positive result for a potential air-spreading pathogen in human beings [12]. Such pathogens are listed in Table 1.

Two kinds of respiratory isolation rooms should be available in every ICU [60]:A negative-pressure room for patients who were colonized or infected by potential air-spreading pathogens (i.e., coronaviruses, Mycobacterium tuberculosis, varicella zoster virus, and measles);A positive-pressure room for patients who are likely more susceptible to acquiring an infection, such as solid-organ transplant (SOT) recipients, hematopoietic stem cell transplant (HSCT) patients, patients with the presence of hematological disorders, and patients with chronic use of corticosteroids, calcineurin inhibitors, anti-metabolites, and other immunosuppressants.

A differential pressure is created when one space (corridor or anteroom) has a different pressure compared to an adjoining space (in this case, the isolation room). When a differential pressure is present, air is forced to flow from the high-pressure space to the lower-pressure space. The direction of the flow is called ‘negative’ when the air ventilation system generates an air flow into the room, but it does not escape from the room; it is positive when the reverse occurs (Figure 1).

The duration of isolation depends on the possibility of pathogen eradication [12].

Whenever the pathogen is eradicated, the patient can leave the isolation room.

A single isolation room is mandatory in the case of some airborne pathogens (tuberculosis, measles, and varicella virus) and patients with neutropenia [7].

### 9.2. Contact Isolation

Contact isolation is required every time a skin or rectal sample (swabs) has a positive result for a potential pathogen transmitted through direct or indirect contact in human beings, especially MDROs (multi-drug resistant organisms) [7]. Those include all pathogens with acquirable resistance through through plasmid transmission, among others, along with ESBL (extended-spectrum beta-lactamase) resistance [7]. Such pathogens are listed in Table 1.

Contact isolation is also required in the case of the diagnosis of particular diseases known for being transmitted by contact (i.e., Ebola).

Contact isolation is mandatory for both patients who are infected and patients who are colonized by these organisms [7,12]. CRE rectal colonization could last for up to one year [61], while VRE lasts for approximately 6 months [62]. MRSA skin colonization has been reported to last for an average of 9 months [63,64,65], and an older age is associated with a longer duration of colonization for both MRSA [63] and CRE.

## 10. MDRO Decolonization

Although the eradication of the multidrug-resistant organism (MDRO) could possibly serve to prevent both further transmission and infection development [66], it is not currently recommended.

### 10.1. Gram-Negative Bacteria (GNB)

There is no recommendation in favor or against routine MDR-GNB decolonization in ICU patients by actual guidelines.

In general patients, the ESCMID-EUCIC guidelines do not recommend the routine decolonization of 3GCephRE and CRE carriers, though they do not extend this statement to immunocompromised patients (e.g., patients in the ICU, with neutropenia, or receiving a transplant) as only few studies have been conducted on this population. Its effectiveness and long-term side effects are encouraged to be assessed through appropriate RCTs (randomized controlled trials) [67].

However, several recent studies suggest an increased risk of CRE infection development in ICU patients with CRE colonization [68,69,70,71] and satisfactory rates of decolonization effectiveness [72,73].

For CRAB (*carbapenem-resistant Acinetobacter baumannii*), AGRE (*aminoglycoside-resistant Enterobacteriaceae*), CoRGNB (*colistin-resistant Gram-negative organisms*), CRSM (*cotrimoxazole-resistant Stenotrophomonas maltophilia*), FQRE (fluoroquinolone-resistant Enterobacteriaceae), PDRGNB (*pan-drug-resistant Gram-negative organisms*), and XDRPA (*extremely drug-resistant Pseudomonas aeruginosa*) carriers, evidence is still limited, and no recommendations have been proposed for ICU carriers nor for non-ICU carriers [67].

### 10.2. Gram Positive Bacteria (GPB)

To our knowledge, MRSA decolonization with intranasal mupirocin and chlorhexidine bathing is not explicitly recommended by any guidelines [15], except for those on an orthopedic or cardio surgery waiting list [74]. Still, there is much evidences that systemic screening followed by the decolonization of MRSA in all ICU patients (universal approach) decreases the incidence of MRSA colonization or infection by up to 52% [75]. In fact, the SHEA/IDSA/APIC guidelines highlight that active surveillance with contact precautions is inferior to universal decolonization in reducing MRSA isolation in adult ICUs [15] (REDUCE MRSA Trial) [76], and universal decolonization with daily CHG bathing plus 5 days of nasal decolonization should be performed in this setting to reduce endemic MRSA clinical cultures [15] (quality of evidence: high). Therefore, the endemic status should be assessed. Predictors of decolonization failure could be respiratory tract colonization [77], a younger age (0–17 years) [66], refugee status [66], and the presence of one or more comorbidities [66]; patients with these factors would possibly need different decolonization strategies.

However, physicians should bear in mind that MRSA colonization is associated with a 4-fold increase in the risk of MRSA infection development [78]. More than 50% of patients with MRSA colonization develop the infection in the ICU setting [79], and MRSA colonization is also associated with an increase in hospital admission, with further consequent possible transmission and outbreak development [33].

As far as we know, no guidelines have been elaborated on VRE decolonization indications or practice. This is probably due to the scarcity of studies conducted on this topic so far. Some studies on MRSA decolonization showed that chlorhexidine bathing could be effective in reducing VRE acquisition and infection development [15]. Cheng et al. obtained VRE decolonization by applying a combination of polyethylene glycol for bowel preparation, a five-day course of oral absorbable linezolid and non-absorbable daptomycin to suppress any remaining VRE, and subsequent oral *Lactobacillus rhamnosus* GG beyond environmental cleaning and isolation [80]. A non-antibiotic decolonization protocol consisting of a four-item bundle for both VRE and CRE was recently proposed by Choi et al.: using a glycerin enema for mechanical evacuation, daily lactobacillus ingestion for the restoration of normal gut flora, a chlorhexidine bath, and changing bed sheets and clothing every day [81]. Both proposed protocols need to be experimented with in further studies to assess their efficacy, but firstly, studies on VRE decolonization benefits should be conducted.

### 10.3. Candida auris

According to the Centers for Disease Control and Prevention (CDC), the efficacy of *Candida auris* decolonization is not known [82]. Chlorhexidine or topical antifungals have been proposed empirically, but evidence is still scarce.

*Candida auris* is currently the biggest emergent threat in USA and European ICUs as, contrarily to other MDROs, no antifungals, single or in combination, have shown solid efficacy. Thus, IPC measures are the best available weapon. Beyond the ECDC’s in-hospital hygiene recommendations, contact tracking, single-room contact isolation, surveillance though periodic skin swab testing of the healthcare personnel, co-hospitalized patients, and cohabitants who came into contact with the *C. auris* carrier could be effective in tackling the spread of *C. auris*.

## 11. Hand Hygiene

Hand hygiene (HH) is crucial for infection control. According to the WHO’s recommendations, hand hygiene should be performed when moving from one patient to another in all settings regardless of the presence of an ongoing infection or colonization.

The WHO’s recommendations on HH are based on two rules: the six movements and the five moments of hand hygiene [59].

Healthcare staff cannot exempt themselves from knowing these rules and should put them into practice as per the strong evidence these practices have shown.

It has been proven that appropriate HH is associated with a reduction in HAI incidence of up to 50% [83], including a 50% reduction in MRSA infections.

Despite the success rate in preventing HAI development and spreading declared by the WHO, a systematic review conducted by Kathryn Ann Lambe and colleagues in 2019 emphasized that the mean HH compliance was only 59.6% in adult ICUs, ranging from 64.4% in high-income countries to 9.1% in low-income countries. This percentage also varies in consideration to the type of ICU (neonatal, 67.0%; pediatric, 41.2%; and adult, 58.2%) and the type of healthcare workers (nurses, 43.4%; physicians, 32.6%; and others, 53.8%) [84].

A Brazilian study estimated that within 20 s without adhering to contact precautions, there was a 45% possibility that HCW (healthcare worker) hands became contaminated with a CRE. After shaking hands with this HCW, the possibility to become contaminated was likewise 22% [85]. If the first HCW had used a gown and gloves or would have washed their hands, the possibility rates would have been, respectively, 10% and 0% [85].

In the case of an outbreak, it would be useful to implement compliance with direct observations of the ‘five moments’ for hand hygiene performed by healthcare workers, followed by individualized verbal feedback [52].

## 12. Shoe and Medical Equipment Hygiene

Shoe soles represent a potential vector for pathogen transmission [86]. In addition to hand-hygiene, HCWs’ shoe bottoms can carry pathogens from one environment to another. Therefore, decontamination is needed when moving from one patient to another, especially when there is an MDRO carrier. Rashid et al. conducted a systematic review looking for an effective decontamination strategy for shoe soles in 2016 but did not succeed. This was also due to the scarcity of data present on this topic. Among mechanical strategies, the use of shoe covers or disposable boots seemed to be the most effective method in reducing the bacterial load in a sanitary setting, while adhesive mats proved to be ineffective [86]. Among chemical strategies, tanks or adhesive mats supplemented with 3–1 benzoisothiazolin or 0.2% benzylkonium were able to reduce the bacterial load [86]. Also, treating boots with a peroxygen disinfectant reduces the bacterial load by up to 1.4 log_10_. Boot baths with 6% sodium hypochlorite seem to prevent virus transmission [87].

On par with HCWs’ shoes, all HCW equipment, including badges, stethoscopes, oximeters, ultrasonography probes, and smartphones, should be disinfected with antiseptics such as chlorhexidine and benzalkonium, although MDR-efflux pump QAC carriers or GNB could be resistant [88].

## 13. Screening

The key points for MDRO screening are summarized in Box 4.

Box 4Key points for multidrug-resistant organism (MDRO) screening.
Use of risk-assessment scores for MDRO acquisition and infection development;Active screening for MDROs through weekly sample collection (skin, rectal, and/or respiratory according to the pathogen);Periodical environmental sample surveillance on high-touch surfaces (no standardized period has been proposed, but it would be advisable to perform this procedure at least once a week);Whole genome sequencing (WGS) whenever an MDRO is identified to identify putative transmission chains and to stratify patients.;Isolation, contact precautions, hand hygiene, and environmental cleaning should be performed in conformity with actual guidelines (see Environmental Cleaning and Hand Hygiene Sections).


a.Risk-assessment scores

Risk-assessment scores (Table 5) could be applied at admission and recalculated daily in order to foresee the risk of colonization acquisition and/or infection. Hereby, HCWs can promptly put into practice the consequential IPC measures.

To our knowledge, no definitive colonization score was elaborated so far for CRAB and CRPA, although Dalben et al. identified some colonization risk factors for their acquisition in ICUs: the male sex, surgery prior to admission, the APACHE II score, and colonization pressure in the week before an outcome [89]. Tacconelli et al. identified some other risk factors for CRAB colonization and infection development, such as quinolone use [90]. Meschiari et al. identified as independent risk factors the use of permanent devices, mechanical ventilation, urinary catheters, the McCabe score, length of stay, and carbapenem use for CRAB colonization acquisition in the ICU setting [91].

**Table 5 antibiotics-13-00789-t005:** Applicable risk-assessment scores for outbreak-related pathogen colonization and/or infection development.

Type of MDRO	Type of Risk	Risk-Assessment Score	Description	Performance
*Candida* spp. [92]	Colonization	*Candida* Colonization Index [93]	Ratio of number of (non-blood) sites colonized with *Candida* spp. /total number of sites cultured Threshold = 0.5	PPV = 66% NPV = 100%
	Infection	*Candida* score [94]	*Candida* Score = TPN (1 point), surgery (1 point), severe sepsis (2 points), multifocal *Candida* colonization (1 point); threshold = 2.5	Sensitivity = 81% Specificity = 74% PPV = 16% NPV = 98%
		Ostrosky-Zeichner Clinical Prediction Rule [95]	Mechanical ventilation ≥48 h AND systemic antibiotic AND CVP (on any day in day 1–3 period of ICU admission) plus ≥1 of the following: any major surgery (days 7–0), pancreatitis (days 7–0), use of steroids/other immunosuppressive agents (days 7–0), use of TPN (days 1–3), or dialysis (days 1–3)	Sensitivity = 50% Specificity = 83% PPV = 10% NPV = 97%
ESBL-producing *Enterobacteriacae*	Colonization	Tumbarello et al. [96]	Recent (≤12 months before admission) hospitalization, transfer from another healthcare facility, Charlson comorbidity score ≥ 4, recent (≤3 months before admission) β-lactam and/or fluoroquinolone treatment, recent urinary catheterization, and age ≥70 years	With cutoff score ≥ 3: Sensitivity = 94% Specificity = 41% PPV = 44% NPV = 93%
	Infection (BSI)	ESBL Prediction Score (ESBL-PS) [97]	Outpatient procedures within 1 month, prior infections or colonization with ESBL-E within 12 months, and number of prior courses of β-lactams and/or fluoroquinolones used within 3 months of BSI	With cutoff score ≥ 1: Sensitivity = 88% Specificity = 77% PPV = 16% NPV = 99% With cutoff score ≥ 3: Sensitivity = 43% Specificity = 96% PPV = 33% NPV = 97%
CPE	Colonization	Papafotiou et al. [98]	Karnofsky score, previous hospitalization, stay in a long-term care facility, history of ≥2 different interventional procedures, previous CPE colonization or infection, renal replacement therapy, and diabetes with end-organ damage	With cutoff score ≥ 27: Sensitivity = 72% Specificity = 81% PPV = 15% NPV = 98%
CRAB	Infection	Cogliati Dezza et al. [99]	CRAB colonization, higher CCI, multisite colonization, and the need for mechanical ventilation	Unknown
XDR *A. baumanii*	Colonization	Moghnieh et al. [100]	Urinary catheter placement >6 days, ICU contact pressure for >4 days, presence of gastrostomy tube, and previous use of carbapenems or piperacillin/tazobactam	Unknown
MRSA	Colonization	Torres et Sampathkumar [101]	Nursing home residence, diabetes, hospitalization in the past year, and chronic skin condition/infection	With cutoff score ≥ 8: Sensitivity = 54% Specificity = 80%
VRE	Colonization	PREVENT score [102]	Age of ≥60 years, hemato-oncological disease, cumulative antibiotic treatment for >4 weeks, and VRE infection	Sensitivity = 82% Specificity = 77% PPV = 57% NPV = 92%
MDROs	Colonization	AutoRAS- MDRO [103]	Electronic health records (EHRs)	Sensitivity = 81% Specificity = 79% PPV = 49% NPV = 94%

BHI: brain heart infusion; CPE: carbapenemase-producing *Enterobacteriaceae*; CRAB: carbapenem-resistant *Acinetobacter baumanii*; CRE: carbapenem-resistant *Enterobacteriaceae*; CR-GNB: carbapenem-resistant Gram-negative bacteria; ESBL: extended-spectrum beta-lactamase; ICU: intensive care unit; IPC: infection prevention and control; HCW: healthcare workers; MDRO: multidrug-resistant organism; MRSA: meticillin-resistant *Staphylococcus aureus*; VRE: vancomycin-resistant *Enterococci*, XDR: extensively drug resistant.

b.CRAB screening

Actually, there is no consensus on CRAB active screening strategies [12]. Garnacho-Montero J et al. recommend weekly rectal, pharyngeal, and tracheal swabs [104]. Valencia-Martìn et al. found a sensitivity of 96% by combining rectal and pharyngeal swabs compared to 78% when using rectal swabs only [105]. Different values but the same conclusion were drawn by Nutman et al.: a 94% sensitivity was obtained when combining buccal mucosa, skin, and rectal swabs compared to 74% when using rectal swabs only [106]. They also found that the most sensible swab was buccal mucosa swab for respiratory culture-positive patients and skin swab for respiratory culture-negative patients. Meschiari et al. found that skin samples (100%), followed by rectal samples (86%), showed the best sensitivity, but due to the waiting period to receive a screening test, they suggested adopting contact precaution measures to all ICU patients until the end of the outbreak [52].

c.Rectal screening for carbapenem-resistant Gram-negative bacteria (CR-GNB)

This screening procedure should be performed at ICU admission and repeated at least once a week according to local epidemiology [12,107]. In order to promptly identify a patient with CR-GNB rectal colonization or CR-GNB infection, an active surveillance system involving the microbiology laboratory and infection control staff should be implemented [108].

Contact precautions should be adopted, including the following [12,107,108]:-Single-use gloves and gowns should be worn during assistance (worn at the moment of entering the room of the patient with CR-GNB colonization and removed at the moment of exiting the patient’s room);-Gloves and gowns should be used individually for every patient with CR-GNB colonization, since the CR-GNB could vary in species and resistance profile;-Gloves and gowns should be changed according to the WHO’s guidelines regarding the ‘Five moments’ and ‘six movements’ [59].d.Skin screening for MRSA

As for rectal screening, skin screening should be performed at admission and repeated at least weekly in the ICU [15]. Other situations in which active screening is encouraged are preoperatively, upon initiating dialysis, at admission to a particular unit, or upon identifying a potential outbreak [15]. Swab samples should be collected from the nostrils, throat, and perineum. Other sites could include the wound, sputum, or eyes [66].

e.Environmental samples surveillance

Environmental samples should be collected according to the CDC Environmental Checklist for Monitoring Terminal Cleaning in order to prevent the spread of CR-GNB and other dangerous microorganisms, paying particular attention to high-touch surfaces [109] (see the Cleaning Section).

Environmental samples should be collected with a sterile BHI moistened gauze, as normal swabs revealed a low sensitivity for *Acinetobacter baumanii* (0 to 18%) [52,110].

f.Whole genome sequencing (WGS)

The genomic characterization of CR-GNB could be useful to identify putative transmission chains [111] and to stratify patients [52]. For instance, lately, non-functional adeN was found to be associated with increased virulence and hyper invasiveness [112]. In Meschiari et al.’s study [52], only two patients who acquired a CRAB clone with the inactivation of adeN survived, probably because they had a younger age and better immune status. Their hypothesis was that the inactivation of adeN could have contributed to higher mortality rates in their outbreak, similarly to other studies [113,114,115], despite appropriate therapy with cefiderocol.

## 14. Environmental Cleaning

Cleaning the room and bed is essential for IPC in the ICU. For this reason, cleaning should be standardized with a hospital protocol and realized on a routine basis or when a patient is moved or discharged from a room (i.e., terminal cleaning). In the protocol, environmental service personnel training, the use of checklists, and/or monitoring ‘high-touch’ contact surfaces with healthcare workers’ hands should be provided [109].

ICU cleaning includes both surface cleaning and air cleaning.

### 14.1. Air Cleaning

A ventilation system, together with the appropriate use of heating and air conditioning, is fundamental in preventing the acquisition of HAIs. High-efficiency particulate air (HEPA) could be useful for the prevention of fungi infections, including Aspergillus spp [116]. Recently, air purifiers seem to be effective in reducing the microbial load in the air and on surfaces in the ICU [117], and it may be worth including them in the ICU cleaning routine.

### 14.2. Surface Cleaning

Cleaning, including cleaning of isolation rooms and open-space areas, should be performed with 10% sodium hypochlorite for environmental surfaces and hydrogen peroxide wipes for all medical devices. This has also proven to be effective against *C. auris* contamination [118,119].

It should be performed on all surfaces, with particular focus on the most ‘high-touch’ surfaces [109], defined by Kisk Huslage et al. in 2015 as sustaining more than three contacts per interaction with the patient [109]. Among the 109 ICU surfaces studied, three were identified as ‘high-touch’ surfaces, namely the bed rail, the bed surface, and the supply cart. These three surfaces accounted for 40.2% of the contacts recorded in the ICUs. Considering the medical–surgical floor, the ‘high-touch’ surfaces, defined as sustaining more than one contact per interaction, were the bed rail, the over-bed table, the intravenous pump, and the bed surface (48.6% of all contacts with medical–surgical floors). In the same study, it was found that bed rails had the highest frequency of contact in both types of healthcare settings, accounting for 7.76 contacts per interaction in the ICUs [109].

In order to write a hospital protocol, a local assessment of the ‘high-touch’ surfaces should be performed and integrated with the above-mentioned data.

Of course, the protocol must take into consideration the concentration and type of pathogens found on the specific environmental surfaces to determine the best kind of disinfection.

Several studies demonstrated that standard cleaning with self-monitoring is insufficient to control CRAB environmental spread [52,120]. This information becomes more relevant considering that environmental contamination seems to be the most frequent source of CRAB cross-transmission in the ICU [52,106,120].

Moreover, Carling PC et al. highlighted that less than 50% of standardized environmental surfaces have been cleaned during terminal room cleaning [121].

The cleaning process should not only follow the CDC’s Environmental Checklist for Monitoring Terminal Cleaning guidelines [122], but also put into practice Meschiari’s *‘cycling radical cleaning and disinfection’* from the *‘five-component bundle’* protocol [52]. Environmental contamination appeared to represent the most frequent source.

Recently, ‘no-touch’ cleaning methods have been developed, including UV cleaning and pressurized hydrogen peroxide. Although they are effective, they do not tend to be well-tolerated and they are expensive and limitedly practical, as it takes hours before the room is ready for a new patient [123,124]. This makes the *cycling radical cleaning and disinfection* method [52] preferable as a faster, easy-to-use, and cost-effective method.

## 15. Antimicrobial Stewardship Program

IPC in the ICU setting is the result of teamwork [125,126,127] and effective communication [128]. Beyond ICU personnel (doctors, nurses, and HCWs), three key roles are needed to perform antimicrobial stewardship: the infectious diseases specialist (IDS) [129], the clinical microbiologist [130], and the clinical pharmacology specialist [131].

In case no protocol has been elaborated at the facility level, the IDS should be consulted at the following times [132]:-Whenever an infectious disease is suspected;-When the patient presents fever;-Whenever a new cultural or serological positivity is released by the microbiological laboratory;-For antimicrobic therapy initiation, monitoring, and discontinuation.

Adherence to IDS recommendations by the treating doctor has been proven to be of paramount importance for disease progression and outcomes, and also in terms of mortality [133,134].

The timing of specialists’ consultation is essential, and a proactive approach compared to an event-triggered approach would be preferrable [129]. In this regard, Zwerwer et al. recently managed to develop a machine-learning model that is able to predict infection-related consultations in ICUs up to eight hours in advance based on electronic health records [129].

The IDS should perform at least the first consultation for every patient at the bedside when visiting the patient [135]. The IDS should visit the patient every time an important clinical change is present. According to the number and severity of patients suffering from bacterial, virological, or fungal infections, a minimum number of weekly visits should be planned [132].

Although many studies have witnessed the commonly inappropriate prescription of antibiotics identified as ‘reserve antibiotics’ in the WHO’s AWaRe antibiotic book [136] worldwide, no exclusivity to IDS prescribers has been established [137].

For hospitals without IDS services, Zimmermann et al. are currently conducting a trial with the purpose to identify means to comprehensively and sustainably improve the quality of care of patients with infectious diseases in those settings (trial registration: DRKS00023710) [138].

Antimicrobial stewardship (AMS) remains pivotal and complementary to IPC in fighting antimicrobial resistance.

## 16. Outbreak Reporting

Manuscripts on IPC are mainly conducted during outbreaks. The main limitation of this kind of study is that it is scarce, and frequently, different risk factors are taken into consideration from one study to another [7].

Another limitation is that a universal outbreak definition is lacking [7,139]. One of the most accurate definition lists for different pathogen outbreaks is the one offered by the Division of Infectious Disease Epidemiology, West Virginia, USA [8].

The ORION statement (Outbreak Reports and Intervention Studies of Nosocomial Infection statement, 2007) by Sheldon Stone and colleagues proposed a standardized way of reporting an outbreak, which could be useful in the prevention and/or management of future outbreaks, other than contributing to the current literature [140].

The statement consisted of a 22-item checklist including information on the number of colonized, infected, and deceased patients; the type of medical department; the number of beds in the ward; performance of genotyping; the study design; and data on costs.

A decade after ORION statement publication, outbreak reports globally still do not provide the basic information in the event [139]. After 2017, only a review on CRAB and CRPRA outbreaks mentioned the statement, apparently not using it for the selection of the outbreak reports, but highlighting the importance of appropriate reporting [141].

## 17. CRE Prevention among Special Populations

### 17.1. Hematological Patients

Among hematological patients with carbapenem-resistant *Enterobacteriaceae* (CRE) rectal colonization, in a recent retrospective study by Xia Chen et al., receiving proton pump inhibitors and admission to the ICU (*p* < 0.05) were identified as risk factors for subsequent CRE infection development [142]. Receiving proton pump inhibitors is also recognized to be a predisposing factor to infection by extended-spectrum β-lactamase-producing Enterobacteriaceae. Among this kind of hematological patients, gastrointestinal injury, tigecycline exposure, and the carbapenem resistance score were not associated with subsequent CRE infection, which may be responsible for subsequent CRE infection in other hematological disorders [143], as well as high-risk disease and mucositis [144].

### 17.2. Neutropenic Patients

According to the ESCMID-EUCIC guidelines, there is no conclusive evidence on 3GCephRE carriers’ decolonization benefits in this population. In particular, the decolonization of 3GCephRE has been associated with temporary effectiveness and an increased risk of developing ESBL-E BSI in patients with neutropenia colonization [67].

For future clinical trials on decolonization by this pathogen, it is suggested to use a combination of oral colistin sulphate (50 mg (salt) four times daily) and neomycin sulphate (250 mg (salt) four times daily) in patients with severe neutropenia [67].

### 17.3. Hemodialysis Patients

Patients using a temporary line for vascular access have a greater risk of colonization by MRSA [32].

## 18. Cost-Effectiveness and MDRO Reproduction Number (R_0_)

The cost-effectiveness of IPC strategy implementation, such as screening, laboratory tools, HCW personnel, and bed rotations (that require one bed off regular admissions) are to be considered.

In the WHO’s 2022 global report, the impact and cost-effectiveness of IPC measures was addressed to encourage the improvement of IPC programs [145].

Multidrug-resistant organisms and difficult-to-treat infections are associated with prolonged hospitalization with higher costs in terms of human resources, assistance, drugs, disposables, additional cleaning, length of stay, and laboratory costs. MDROs’ basic reproduction number (R_0_) should be kept in mind when estimating an outbreak cost (Table 6). In fact, the basic reproduction number (R_0_), or basic reproductive number, expresses how many successful transmission events and new infections result on average from one infection [146].

### 18.1. CRE

Lin et al. developed a computational model in order to predict the cost-effectiveness of CRE surveillance strategies in the ICU [152]. The cost of a single patient affected by CRE was estimated to be USD 63,948 based on a literature review. Other than reducing CRE colonization acquisition, they found out that up to USD 572,000 per year could have been saved whenever IPC strategies were implemented in Maryland, USA considering Maryland’s 2012 incidence of 4.8 CRE infections in every 100,000 persons.

The rate of CRE has risen exponentially.

A single identification of a CRE infection or colonization could be responsible for up to 11 transmissions according to a Brazilian study [85].

In this study, the authors developed a mathematical model to describe the dynamics of transmission of CRE in the ICU, and they found the CRE transmission R_0_ (*basic reproduction number*) to be 11 with routine IPC before the implementation of the experimented IPC strategies they performed. After the IPC implementations, the R_0_ dropped to 0.41 (range 0–2.1). To our knowledge, this is the only study that was capable of estimating the R_0_ of patients with CRE colonization.

Recently, many effective new antibiotics have been discovered against CRE [153], but their costs are still very high.

### 18.2. VRE

Mac et al. proved the same cost-effectiveness of VRE screening and isolation in a medicine ward in Canada [154]. The cost of a single VRE patient was esteemed to be USD 17,949 [155], while a VRE outbreak costed EUR 60,524 [156] based on a literature review. Similar to Lin et al., for CRE, they achieved both VRE colonization acquisition and relative mortality reduction at a cost-effectiveness threshold of 50,000 USD/QALY (*quality-adjusted life years*) in Toronto, Canada. According to current studies in the literature, the VRE transmission R₀ was 1.32 (range of 1.03–1.46) [157].

### 18.3. MRSA

Chaix et al. estimated the cost of a single MRSA infection in a French ICU to be USD 9275, while the IPC measures for MRSA would range from USD 340 to USD 1,480 per patient and USD 30,225 for the entire outbreak [158,159]. They proved that routinary screening with other IPC measures managed to reduce both costs and MRSA incidence by 14%. According to a recent review, universal decolonization would be more effective and less expensive than other IPC strategies, but the most effective would be a combination of screening, isolation, and decolonization in the ICU setting, even though it is the most expensive one [79]. Eike Steinig and colleagues conducted the first study on the community-acquired MRSA R_0_, resulting in a range between 0.97 and 1.60 depending on the strain [147].

In summary, IPC measures in the ICU have been proven to be cost-effective wherever MRSA colonization and infection rates are significant, although no cut-off rate has been assessed.

### 18.4. CRAB

The literature on carbapenem-resistant *Acinetobacter baumanii* (CRAB) outbreak costs is more scarce. Coyle et al. elaborated on a model estimating that a single patient with CRAB would cost up to USD 55,122 for a 13-day length of stay [160], confirmed by Young et al., who reported a real-life data cost of UAS 60,000 in a Korean ICU [38]. Considering an R_0_ of approximately 1.5 in Australian ICUs, the total outbreak cost would be around USD 1 million [28]. Implementing IPC measures, the threshold would be of $75,000–$93,822/QALY.

### 18.5. Candida auris

Taori et al. analyzed the cost a *Candida auris* outbreak in London, UK, which was estimated to be EUR 1,217,817. The additional length of stay accounted for half of this sum (EUR 69,645 per month) [161]. The screening cost for *C. auris* was estimated to be EUR 269,984 during the outbreak (EUR 51,040 per month) [161].

Considering this study, a *C. auris* outbreak exceeds in costs an average CRE outbreak (EUR 1.1 million) [162] and a *Clostridium difficile* outbreak (EUR 1,222,376) [148], taking into consideration the long-lasting contamination or the need to close the ICU for a certain period.

The cost-effectiveness of IPC measures in *C. auris* outbreaks still needs to be assessed. Recently, Rosa et al. managed to prove the positive economic impact of the implementation of an in-house PCR (polymerase chain reaction) to screen patients presenting risk factors for *C. auris* acquisition at admission in Miami hospitals in the USA [163]. The saving margin in a two-year post-intervention period was between USD 77,251,310 and USD 373,048,026 based on a deduced incidence of positivity of 3% [163].

As far as we know, no *C. auris* studies conducted so far identified the *C. auris* transmission R_0_.

Therefore, IPC represents a solid cost-effective solution for CRE, VRE, MRSA, and CDI outbreaks and a possibly cost-effective strategy for *C. auris* outbreaks, as it seems to be capable of preventing these hospitalizations with associated costs.

The reproduction numbers of other pathogens possibly responsible for outbreaks in the ICU are reported in Table 7.

## 19. New Experimental Strategies

Beyond HH and isolation precautions, new experimental IPC strategies have been proposed in the past 10 years. These strategies are focused on MDRO outbreaks (Table 8).

The most recent applications include the employment of artificial intelligence and machine learning, but the literature on this topic is still scarce.

One of the most relevant, easy-to-implement, and effective strategies is the five-item IPC bundle proposed by Meschiari et al. for CRAB outbreaks in the ICU (Box 5) [52]. Notably, A. baumannii stands out for its endurance, and it could survive on dry surfaces for up to 5 months [180], facilitating its spreading.

Box 5Meschiari’s ‘five-component bundle’ of IPC57.
Proactive reinforcement of all routine IPC practices among HCWs:Improving HH compliance with direct observations of the WHO’s ‘5 moments’ for HH performed by IPC nurses followed by individualized verbal feedback;Establishing an ‘improvement group’ with medical and nursing staff to analyze critical issues regarding HH compliance;Monitoring compliance with contact precautions performed by IPC nurses using two specific checklists;Meetings with radiology and transport personnel to reinforce compliance with IPC measures.Extended CRAB screening: For all patients with an expected ICU length of stay > 24 h in ICU;At admission and weekly thereafter;Swabs should be performed in axilla, groin, trachea, and rectum. Contact precaution measures:For all patients until discharge, independently of CRAB status;Single-use gloves and gowns should be worn before entering each single-patient unit, and gloves should be changed according to the *WHO’s 5 moments for HH* [59]).Environmental sampling:By means of pre-moistened sterile gauze pads, suggested by Corbella et al., as it showed an increased sensitivity for *A. baumanii* [181]; After rubbing all ICU surfaces vigorously, moistened gauze pads should be firstly put in incubation for 24 h at 37 °C in a screw-cap container with 10 mL of brain heart infusion medium (BHI), and secondly sampled into MacConkey agar plates and incubated aerobically at 37 °C for 48 h. Cycling radical cleaning and disinfection:Performed for all rooms, common areas, and patients;Use of 10% sodium hypochlorite for environmental surfaces;Use of hydrogen peroxide in wipes for medical devices;Cleaning and disinfection should be performed from upper corner to opposite lower corner starting from a transitory unit, and disinfection should be checked by IPC nurses through fluorescein spray with an UV torch, with special attention to hard-to-reach areas, and if not effectively cleaned, disinfection should be repeated; All disinfected surfaces should dry completely before re-using them;After common area sanitization, the colonized patient is moved from their original unit to a transitory unit, where the patient’s disinfection is performed with 2% leave-on chlorhexidine disposable cloths while the patient’s original unit becomes disinfected, and once the bed is cleaned, the disinfected patient can come back to their original room, and the transitory room becomes cleaned thereafter;The whole cycle process takes around 6 h to be completed;The process requires the recruitment of two people dedicated to cleaning and an additional nursing shift.


Marianna Meschiari et al., when facing a CRAB outbreak in their ICU, decided to implement and systematize IPC measures, which led to the elaboration of this successful protocol (Box 4).

While previously existing items n. 2 and n. 3 were intensified and revised (multiple sites vs. rectal site for n. 2 and universal vs. CRAB carrier-only contact precaution measures for n. 3), items 4 and 5 are novelty in the field. In their study, a whole genome sequencing (WGS) analysis was performed for all CRAB isolates, environmental or clinical.

The pitfall of this new method is that it is frequently difficult to create a ‘transitory room’ due to ICU overcrowding currently affecting many ICUs all over the world [182].

Moreover, the whole process takes approximately 6 h, which implies the need for supplementary HCWs or, more realistically, healthcare assistant shifts, contributing to work overload [183]. It could be still useful to avoid ICU closure and limiting admissions due to extensive CRAB contamination. It is also applicable to open-space ICUs, the ICUs most affected by nosocomial epidemics [184]. After the introduction of *cycling radical cleaning and disinfection* in 2018, the Modena ICU (Italy) did not experience nosocomial ICU CRAB outbreaks anymore, but only sporadic cases [52]. Furthermore, ICU alcohol hand rub use increased by more than three times, and total antibiotic use dropped by 18.2%, while meropenem and fluoroquinolones dropped by 83.3% and 84%, respectively (the percentages were calculated based on the original article’s data).

## 20. Future Perspectives of IPC

It has not escaped our notice that IPC strategies could consistently change in the next few years (Table 9). Phage therapy, targeting specific virulence genes and non-antibiotic decolonization strategies, seem the most promising ones.

Take-home messages are displayed in Box 6.

Box 6Take-home messages.
The most effective IPC strategy remains unknown, as a multimodal approach does not identify the most effective strategy, given that all strategies are applied simultaneously [105].Any outbreak should be a reason to intensify IPC (*infection prevention and control*) measures [52].A single CRE patient could be responsible for up to 11 contagions.Further studies are needed to strengthen ideas in favor or against MDRO decolonization in the ICU setting. A standardized, universal, pragmatic protocol for HCW education should be elaborated. A rapid outbreak recognition tool (i.e., an easy-to-use mathematical model) should be proposed to improve early diagnosis and prevent spreading. Standard cleaning with self-monitoring is insufficient to control MDRO environmental spread.The mechanical removal of biofilm may be more relevant than the type of disinfectant [52,192].Weekly rectal, pharyngeal, and tracheal swabs for CR-GNB should be performed [104].Environmental samples should be collected with a sterile BHI moistened gauze, as normal swabs revealed a low sensitivity for some pathogens like *Acinetobacter baumanii* (0 to 18%) [52,110]IPC strategies proved their cost-effectiveness independently to the country, pathogen, or type of strategy.New promising strategies are emerging and need to be tested in the field.


## 21. Limitations

The limitations of this review include the narrative nature of this study, which led to subjectivity in article and guideline selection. This implies that we might have unconsciously overlooked other variables impacting IPC in our review. We decided to mention some of the preventive measures and risk factors among a few special populations to increase physicians’ attention towards these categories. A deep focus would require a separate dedicated, population-based review.

Considering cost-effective reporting, studies have been conducted in different countries, and the R_0_ should have been calculated differently in each setting; therefore, these numbers could not be equally applicable to every country or setting.

## 22. Conclusions

The lack of IPC strategies or its application have made and still make hospitals, and ICUs in particular, responsible for an increase in MDRO reservoir into the community [7].

Despite the great number of studies on IPC, it is still difficult to evaluate which is the most effective strategy because of intrinsic study limitations [193].

It would be surely interesting to see if Meschiari’s ‘five-bundle protocol’ for CRAB outbreaks could be applied to other difficult-to-control critical pathogen outbreaks, such as CRE and *Candida auris*.

A univocal, numeric, and easy-to-calculate definition of a ‘hospital outbreak’ of a certain infective disease is still lacking. This would accelerate the outbreak identification process by healthcare personnel and promptly put in place IPC strategies. Further studies based on the proposed mathematical model provided by the Brazilian group of Sao Paulo should be encouraged to assess the in-hospital-acquired pathogen R_0_ [85].

Hopefully, in the future, plasmid modifications by genetic engineering would represent a plot twist in CRE infection control strategies as well as phage therapies [185].

## Figures and Tables

**Figure 1 antibiotics-13-00789-f001:**
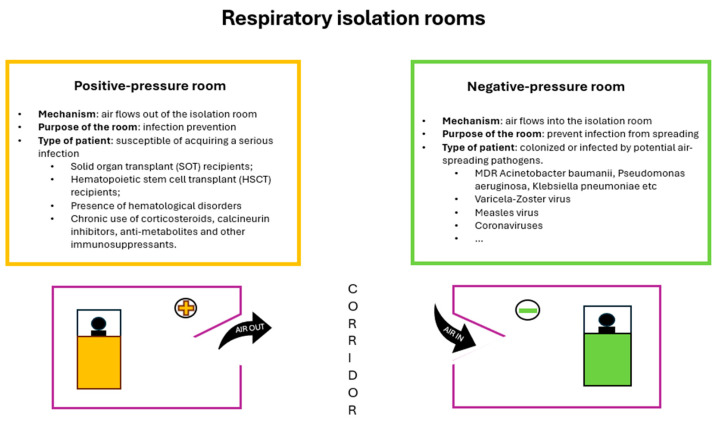
Diagram representing positive-pressure and negative-pressure respiratory isolation rooms.

**Table 1 antibiotics-13-00789-t001:** The routes of transmission for some of the most frequent difficult-to-treat pathogens in the ICU setting.

Routes of Transmission in ICUs
Direct or indirect contact Bacteria: MDR-GNB [12] (including *CRE*, *ESBL carriers* [13], *MDR-Klebsiella* spp., *MDR-Acinetobacter baumanii* [14], and *MDR-Pseudomonas aeruginosa)*, *MRSA* [15], *VRE* [16], and *Clostridium difficile* [16] Fungi: *Candida auris* [17] and *Scedosporium* spp. [18] Virus: Ebola virus [19]
Water contamination [20] Bacteria: *Legionella* spp., *Pseudomonas* spp., *Acinetobacter spp*, and *Serratia* Fungi: *Aspergillus* spp., *Mucor* spp., *Trichosporon* spp., *Scedosporium* spp. [21], and *Fusarium* Virus: *Norovirus*
Air contamination Bacteria [12]: *CRE*, *Acinetobacter baumanii*, *Pseudomonas aeruginosa*, *Corynebacterium* *striatum **, *Legionella* spp., and *MRSA* [22] Fungi: *Aspergillus* spp. [23], *Fusarium* spp. [21], *Scedosporium* spp. [18,21], and *Lomentospora* spp. [21] Virus: *human coronaviruses* (including *SARS-CoV-2* [24]) and Ebola virus [19]
Droplet-spread and airborne infections [25] Bacteria: *Mycobacterium tuberculosis*, *Bordetella* spp., and *Pertussis* Virus: *human coronaviruses* (including *SARS-CoV-2* [24]), *Varicella zoster virus,* *measles virus*, *influenza viruses* (including *H1N1*, *H2N3*, *H5N1*), *parainfluenza* *viruses, respiratory syncytial virus*, and *adenoviruses*

MDR-GNB: multidrug-resistant Gram-negative bacteria; CRE: Carbapenem-resistant Enterobacterales; ESBL: extended-spectrum beta-lactamase. * It did not escape our attention that MDR-Corynebacterium striatum is an emerging MDR-GB (multidrug-resistant Gram-positive bacterium) [26] that soon could be included among pathogens requiring isolation. Although the route of transmission has not been clearly identified [27], as they are frequently presented as respiratory infections having similar characteristics to MDR-GN bacteria, we decided to list it under the ‘Air contamination’ pathogenscategory.

**Table 2 antibiotics-13-00789-t002:** Studies aiming to evaluate the optimal patient-to-nurse (PNR) ratio in the ICU setting.

Type of Study	Study Author and Year of Publishing	Country	Time Period	Sample Size	Suggested ICU PNR	Higher Ratios Were Associated with Higher Mortality
Guidelines	Bray K et al., 2010 (the British Association of Critical Care Nurses, the Critical Care Networks National Nurse Leads) [39]	UK	-	-	1:1	Yes
	American Nurses Association (ANA) and California Legislation (Assembly Bill No. 394)	California, USA	-	-	2:1	Yes
Narrative review	Suresh K. Sharma et Ritu Rani 2020 [40]	India	-	-	1:1	Yes
Retrospective observational study	Falk AC 2023 [41]	Sweden	15 years	2 ICUs (9814 patients)	1:1	Yes
Cross-sectional, retrospective, risk-adjusted observational study	West E et al., 2014 [36]	UK	16 years	65 ICUs (38,168 patients)	0.5:1	Yes *

* ICU with 1:1 compared to 0.5:1 PNR showed a slightly higher estimated mortality in first group, although not statistically significant. Authors conclude that healthcare managers should consider not only PNR, but also their educational level and specific skills. ICU: intensive care unit; PNR: patient-to-nurse ratio; UK: United Kingdom; USA: United States of America.

**Table 3 antibiotics-13-00789-t003:** Studies evaluating the association between the intensivist-to-patient ratio and higher mortality rates in the ICU setting.

Study Author and Year of Publication	Country	Type of Study	Time Period	Sample Size	Median PIR	Higher Ratios Were Associated with Higher Mortality
Neuraz A et al., 2015 [42]	France	Multicenter observational study	2013	5718 patients (8 ICUs)	5.6	Yes
Gershengorn HB et al., 2017 [43]	UK	Retrospective cohort study	2010–2013	49,686 patients (94 ICUs)	8.5	Yes
Dara SI et al., 2005 [44]	USA	Retrospective cohort study	2001–2003	2492 patients (1 ICU)	8.4 *	No
Gershengorn HB et al., 2022 [45]	Australia and New Zealand	Retrospective cohort study	2016–2019	27,380 patients (67 ICUs) in the ‘narrow cohort’ and 91,206 patients (73 ICUs) in the ‘broad cohort’	10.1	No
Agarwal A et al., 2022 [46]	USA	Cross-sectional observational study	2020–2021	1322 patients (62 ICUs)	12	No
Kahn JM et al., 2023 [47]	USA	Retrospective cohort study	2018–2020	51,656 patients (29 ICUs)	11.8	No
Estenssoro E et al., 2017 [48]	Latin America (51% from Brazil, 17% Chile, 13% Argentina, 6% Ecuador, 5% Uruguay, 3% Colombia, and 5% between Mexico, Peru, and Paraguay)	Cross-sectional observational study	2015–2016	257 ICUs	1:1–1:3 (11%) 1:4 to 1:7 (46%) >8 (41%)	Not evaluated

* Calculated. ICU: intensive care unit; PIR: patient-to-intensivist ratio; UK: United Kingdom; USA: United States of America.

**Table 4 antibiotics-13-00789-t004:** Education models proposed to cope with healthcare-associated infections (HCAI) in intensive care units (ICUs).

Study Author and Year of Publication	Country	Type of Infection	Most Relevant Proposed Solutions
Menegueti MG et al., 2019 [51]	Brazil	CAUTI	- Daily checklist for CAUTIs - Biannual training for HCWs
McNett et al., 2020 [55]	USA	VAP	- Educational meetings, auditing, and feedback - Developing tools - Using local opinion leaders - Financial incentives [56] - Building a coalition
Mogyoródi et al., 2023 [37]	Hungary	VAP	- A single educational session - Interactive slide presentation and discussion for nurses and nurse assistants - Education focused on the following: ○ Incidence, risk factors, and pathophysiology of VAP; ○ Recommended preventive measures; ○ Impact of ICU nurses’ compliance on patient outcomes - Elaboration of a poster summarizing the five-component bundle - Two-week duration of educational intervention - Refresh IPC course after one year
Phan et al., 2018 [57]	Vietnam	All HCAI	- Two sessions of 3 h educational course - Focus on HH - Educational program activities: ○ Ten-minute video displaying the reasons for HH ○ Small-group discussion about the reasons for HH ○ Role-playing game where HCWs have to identify pathogens using ultraviolet light on participants’ hands to determine whether their hands had been washed ○ Small-group discussion to determine five moments of HH [58] ○ Practice and discussion of procedural aspects of handwashing technique (six steps of HH) [59] ○ Lecture about efficacy of alcohol-based hand-rub compared to handwashing with water and soap
Moghnieh et al., 2023 [54]	Eastern Mediterranean Region (Afghanistan, Barhain, Iraq, Kuwait, Jordan, Lebanon, Oman, Pakistan, Palestine, Qatar, Sudan, Syria, United Arab Emirates, and Yemen)	All HCAI	- Different training programs for different categories of HCWs - Training outside of the country and exposure to other experienced countries - Strengthening and creating focused IPC education modules in undergraduate health sciences majors - University specialties and higher degrees in IPC - Make IPC training mandatory for all HCWs with periodic licensing and relicensing - Training of hospital administrators - National supervision for IPC education and training - Develop undergraduate IPC education modules - Need for national IPC curriculum

CAUTI: catheter-associated urinary tract infection. HCAI: healthcare-associated infection. HCWs: healthcare workers. VAP: ventilator-associated pneumonia. HH: hand hygiene.

**Table 6 antibiotics-13-00789-t006:** Estimated basic reproduction number (R_0_), single patient, and associated outbreak costs of the most common pathogens responsible for outbreaks in ICU based on research on the literature.

Type of Pathogen	Estimated Mean Single-Patient Cost per Hospital Length of Stay	Estimated Mean R₀	Estimated Mean Outbreak Cost	IPC Implementation Threshold (up to)
CRE	USD 63,948	11	EUR 1.1 million	USD 572,000
CRAB	USD 55,122–USD 60,000	1.5	EUR 1.0 million	75,000–93,822 USD/QALY
VRE	USD 17,949	1.32	EUR 60.524	50.000 USD/QALY
MRSA	USD 9.275	0.97–1.6 [147]	USD 30.225	USD 9.275
*C. auris*	EUR 35.818 *	Unknown	EUR 1.2 million	USD 373,048,026
(HO-CDI)	USD 30,049–USD 34,149 [148,149]	0.55–7.0 [150]	EUR 1.2 million	150,000 USD/QALY [151]

*C. auris*: *Candida auris*; CRAB: carbapenem-resistant *Acinetobacter baumanii*; CRE: carbapenem-resistant *Enterobacteriaceae*; HO-CDI: hospital-acquired *Clostridium difficile* infection. MRSA: meticillin-resistant *Staphylococcus aureus*; QALY: quality-adjusted life year. VRE: vancomycin-resistant *Enterococci*. * Calculated by dividing the total cost of the outbreak by the number of patients involved.

**Table 7 antibiotics-13-00789-t007:** Estimated basic reproduction numbers (R_0_) of other pathogens possibly responsible for outbreaks in the ICU.

Type of Transmission	Type of Pathogen	Estimated R₀ (Mean)	Country	References
Airborne	SARS-CoV-2	1.4 to 6.7 (4.1)	China, Italy, Korea, Peru	[164]
	SARS virus	1.7 to 1.9 (1.8)	Hong Kong	[165]
	MERS virus	2.0 to 6.7 (4.4)	Saudi Arabia	[166]
	H1N1	1.9	China	[167]
	Mycobacterium tuberculosis (MTB)	0.8 to 1.2 0.2 to 0.4 (0.29)	USA	[168,169]
	Measles virus	0.7 to 25.3 (13) * 12–18 (15)	USA, Italy, Japan Systemic review	[146,165,170]
Vectorborne	Zika virus	2.3 to 27.2 (14.9)	Brazil, Chile	[171,172]
	Dengue virus	1.1–1.7 (1.4)	Indonesia, Brazil	[171,173,174]
Bloodborne/ Bodily fluid contact	Ebola virus	1.1 to 10 (1.95)	West Africa	[175]
	HIV (viremic)	(36.8)	Uganda	[176]

* The most recent studies suggest R_0_ < 1 thanks to vaccination.

**Table 8 antibiotics-13-00789-t008:** New experimental strategies in the past 10 years to prevent MDRO spread.

Study Author and Year of Publication	Country	Study Design	Pathogen	Experimental Period	Name of New Strategies
De Freitas DalBen et al., 2016 [85]	Brazil	Prospective study	CRE	Baseline period: 10 months Intervention period: 24 weeks	Educational model based on the following: Simulations of IPC; Weekly auditing and feedbacks; Weekly compliance rates presented in a poster in the unit.
Stachel et al., 2017 [177]	USA	Prospective study	MDROs	8 months	Automated surveillance system to detect hospital outbreak.
Fitzpatrick et al., 2020 [178]	Ireland	Narrative review	All pathogens	-	Artificial intelligence in IPC: driven by ‘big data’, it could find correlations that may indicate medically relevant conditions or identify potential risk factors for outbreaks.
Meschiari et al., 2021 [52]	Italy	Prospective study	*CRAB*	6 years (2013–2019)	Cycling radical cleaning and disinfection.
Piaggio et al., 2023 [179]	Italy	Systemic review	All pathogens	-	Use of smart environments and robots to implement Health 4.0, which is based on the integration of the Internet of Things, Cloud and Fog Computing, and Big Data. HCW awareness and training with respect to the design and use of healthcare technologies that could impact daily work.
Zwerwer et al., 2024 [129]	Netherlands	Prospective study	All pathogens	3 years (2014–2017)	Machine learning model to predict the need for infection-related consultations in the ICU.

ICU: intensive care unit; IPC: infection prevention and control; USA: United States of America.

**Table 9 antibiotics-13-00789-t009:** Proposed future strategies tackling MDRO spread in ICUs.

First Author and Year of publication	Country	Target Pathogen	Aim of the Study	Suggested Technique
Hatfull GF et al., 2022 [185]	USA	MDRB	Fighting antibiotic resistance	Phage therapy
Wang J et al., 2024 [186]	China	MDR-*Corynebacterium striatum*	Fighting antibiotic resistance	Phage therapy
Skurnik et al., 2016 [187]	USA	CPE	Vaccine against CPE (including *NDM-producers E. coli*, *E. cloacae*, *K. pneumoniae*, *K. pneumoniae carbapenemase (KPC)-producing and PNAG-producing P. aeruginosa*)	Vaccine targeting *polysaccharide poly-(β-1,6)-N-acetyl glucosamine* (PNAG) in CPE
Kalfopoulou et Huebner., 2020 [188]	Germany	VRE	Vaccine against *Enterococci* and VRE	Vaccine targeting capsular polysaccharides and surface-associated proteins in *Enterococci*
Miller et al., 2020 [189]	USA	MRSA	Vaccine against MRSA	Vaccine targeting *superantigens* and *pore-forming toxins* in MRSA
Meschiari et al., 2021 [52]	Italy	CRAB	IPC measures in CRAB outbreaks	Targeting *inactivated adeN* gene in CRAB
Ji Yun Bae et al., 2023 [190]	Korea	CRAB	Identifying virulent CRAB genes associated with higher mortality in VAP	Targeting *hisF* and *uspA* genes in CRAB
Choi et al., 2022 [81]	South Korea	VRE and CRE	New non-antibiotic decolonization strategy	4-item bundle: (1) Using a glycerin enema for mechanical evacuation; (2) Daily lactobacillus ingestion for restoration of normal gut flora; (3) Skin cleaning with chlorhexidine; (4) Changing bed sheets and clothing every day.
Wong et al., 2023 [191]	USA	All pathogens	Use of artificial intelligence for new anti infective drug discovery, pathogen pathophysiology and transmission understanding, and diagnostics	Artificial intelligence implementation
Zwerwer et al., 2024 [129]	Netherlands	All pathogens	Using a machine-learning model to predict the need for infection-related consultations in ICU	Machine learning model

CRAB: carbapenem-resistant *Acinetobacter baumanii*; CRE: carbapenem-resistant *Enterobacteriaceae*; CPE: carbapenem-producing *Enterobacteriaceae*. ICU: intensive care unit. MRSA: meticillin-resistant *Staphylococcus aureus*; PNAG: polysaccharide poly-(β-1,6)-N-acetyl glucosamine. VRE: vancomycin-resistant *Enterococci*.

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
