# Peer review of "Infection Prevention and Control Strategies According to the Type of Multidrug-Resistant Bacteria and Candida auris in Intensive Care Units: A Pragmatic Resume including Pathogens R0 and a Cost-Effectiveness Analysis"

_antibiotics, 2024, doi:10.3390/antibiotics13080789_

Round 1

Reviewer 1 Report

Comments and Suggestions for Authors

As hospital acquired infections (HAI) are a major concern of icu and public health, the authors of this paper aimed to provide a pragmatic, physician practice-oriented, resume of strategies towards different MRDOs outbreaks in ICU.

“According to WHO Global Report on Infection Prevention and Control (IPC) of 2022, 7% in high-income countries and 15% in low- and middle-income countries (LMICs) of patients acquire at least one HAI during hospitalization” could authors try to make a definition on what is high and what is low?

Bacteria names such as Klebsiella pneumoniae and Pseudomonas aeruginosa should be in italic type.

MRDOs decolonization, Ct from MRDOs and MDROs?

Why NURSE-TO-PATIENT RATIO is associated with IPC strategies?

Please make an explanation on association between intensivist-to-patient ratio and higher mortality rates in ICU setting.

How those Education models play roles on decreasing healthcare-associated infections (HCAI)?

There are three kinds of isolation: contact isolation, respiratory isolation, or both. The kind of isolation that should be adopted varies depending on where the pathogen was isolated. Could authors make those sentence more detail?

Two kinds of respiratory isolation room should be available in every ICU. If a diagram for this paragraph is prepared, the content will be more readable.

On a par with HCW shoes, all HCW equipment including badges, stethoscopes, oximeters, ultrasonography probe, but also smartphones, should be disinfected with antiseptics such as chlorhexidine and benzalkonium, although MDR-efflux pump QAC carriers or GNB could be resistant” this paragraph should be in an isolated section not in the section of shoes.

Could authors make a comparation on the methods for “SHOE HYGIENE (SH)”.

Authors should give an explanation on what is REPRODUCTIVE NUMBER for R0, which may make the reader clear on this term.

Comments on the Quality of English Language

Moderate editing of English language required

Author Response

Response to Reviewer 1 Comments

1. Summary

Thank you very much for taking the time to review this manuscript. The manuscript has been revised in line with the recommendations of the reviewers. We hope having fulfilled the expectations set by the referees regarding the revisions they requested. Please find the detailed responses below and the corresponding revisions/corrections highlighted/in track changes in the re-submitted files.

Kind regards,

Chiara Fanelli

PhD student in Intensive Care & MD specialist in Infectious Diseases

Department of Medicine, Surgery and Pharmacy, Sassari, Italy

University of Sassari, Sassari, Italy

2. Questions for General Evaluation

Reviewer’s Evaluation

Response and Revisions

Is the work a significant contribution to the field?         

Is the work well organized and comprehensively described?   

Is the work scientifically sound and not misleading?          

Are there appropriate and adequate references to related and previous work? 

Is the English used correct and readable?        

4/5

4/5

4/5

4/5

4/5

Thank you for the positive feedback. I hope that the modifications we apported meet your expectations.

3. Point-by-point response to Comments 

Comments 1: As hospital acquired infections (HAI) are a major concern of icu and public health, the authors of this paper aimed to provide a pragmatic, physician practice-oriented, resume of strategies towards different MRDOs outbreaks in ICU.

“According to WHO Global Report on Infection Prevention and Control (IPC) of 2022, 7% in high-income countries and 15% in low- and middle-income countries (LMICs) of patients acquire at least one HAI during hospitalization” could authors try to make a definition on what is high and what is low?

Response 1: Thank you for pointing this out. It would be certainly better to underline that high/middle/low-income countries’ definitions correspond to World Bank’s income classification determined by the country's gross national income (GNI) per capita. This classification has been adopted by WHO when stating the Global Report on Infection Prevention and Control (IPC) of 2022. In that year, country income classification can be found in this link https://blogs.worldbank.org/en/opendata/new-world-bank-country-classifications-income-level-2022-2023. I therefore added a relative bibliography reference to the sentence.

Comments 2: Bacteria names such as Klebsiella pneumoniae and Pseudomonas aeruginosa should be in italic type. “MRDOs decolonization, Ct from MRDOs and MDROs?

Response 2: Thank for identifying these typos. We found them and adjusted them.

Comments 3: Why “NURSE-TO-PATIENT RATIO” is associated with IPC strategies?Please make an explanation on association between intensivist-to-patient ratio and higher mortality rates in ICU setting.

How those Education models play roles on decreasing healthcare-associated infections (HCAI)?

Response 3: Thank for pointing this out. In consideration to your valid suggestion, we implemented those paragraphs:

L144-152. “The connection between nurse- and intensivist-to-patient ratios and infection prevention and control (IPC) is of paramount importance. Hospital IPC is a sanitary problem involving the whole hospital structure. In fact, IPC strategies are not only for first-line healthcare personnel’s utility, but also to healthcare management personnel’s. Isolation, decolonization, screening and hygiene practices, outbreak reporting may not be implemented or implemented incorrectly if the healthcare staff is understaffed or not properly acquainted with IPC and healthcare-associated infections (HCAI). Thus, resulting in a lack of infection control, beyond a lack infection prevention, with associated costs in terms of human lives and economic expenditure.”.

In this review, we provided guidelines and studies showing how a lower nurse- and intensivist-ratios are associated to a higher mortality in ICU setting (considering infections a major cause of mortality in ICU as per WHO global report on IPC).

L225-227. “In addition to implementing proper nurse- and intensivist-ratios, an educational model on IPC should be adopted by the institution, in order to assure the good quality of IPC strategies implementation.”

Comments 4: “There are three kinds of isolation: contact isolation, respiratory isolation, or both. The kind of isolation that should be adopted varies depending on where the pathogen was isolated.” Could authors make those sentence more detail?

Response 4: Those sentences are meant to be introductive to the following subparagraphs, where the kinds of isolation are described thoroughly: what kind of sample require what kind of isolation, isolation characteristics and what are most common pathogens requiring what kind of isolation (as reference to Table 1). So all details are provided in the next pages.

Comments 5: “Two kinds of respiratory isolation room should be available in every ICU”. If a diagram for this paragraph is prepared, the content will be more readable”.

Response 5: Thank you for the suggestion. We created a diagram for showing in details the two kinds of respiratory isolation characteristics.

Comments 6: “On a par with HCW shoes, all HCW equipment including badges, stethoscopes, oximeters, ultrasonography probe, but also smartphones, should be disinfected with antiseptics such as chlorhexidine and benzalkonium, although MDR-efflux pump QAC carriers or GNB could be resistant” this paragraph should be in an isolated section not in the section of shoes”.

Response 6: Thank you for your suggestion. In order to cope with this issue, we decided to rename the paragraph as ‘SHOE AND MEDICAL EQUIPMENT HYGIENE’.

Comments 7: “Could authors make a comparation on the methods for “SHOE HYGIENE (SH)”.

Response 7: We are sorry, but we were not able to understand what exactly the reviewer was suggesting with this sentence.

Comments 8: “Authors should give an explanation on what is REPRODUCTIVE NUMBER for R0, which may make the reader clear on this term.

Response 8: Thank you for the valuable suggestion. We decided to implement the text as following:

L606-609. “In fact, the basic reproduction number (R0), or basic reproductive number, express how many successful transmission events and new infections result on average from one infection”.

Reviewer 2 Report

Comments and Suggestions for Authors

Please review the attached file.

Comments on the Quality of English Language

Author Response

Response to Reviewer 2 Comments

1. Summary

Thank you very much for taking the time to review this manuscript. The manuscript has been revised in line with the recommendations of the reviewers. We hope having fulfilled the expectations set by the referees regarding the revisions they requested. Please find the detailed responses below and the corresponding revisions/corrections highlighted/in track changes in the re-submitted files.

Kind regards,

Chiara Fanelli

PhD student in Intensive Care & MD specialist in Infectious Diseases

Department of Medicine, Surgery and Pharmacy, Sassari, Italy

University of Sassari, Sassari, Italy

2. Questions for General Evaluation

Reviewer’s Evaluation

Response and Revisions

Is the work a significant contribution to the field?         

Is the work well organized and comprehensively described?   

Is the work scientifically sound and not misleading?    

Are there appropriate and adequate references to related and previous work?    

Is the English used correct and readable?        

4/5

2/5

3/5

4/5

3/5

Thank you for the positive feedback. I hope that the modifications we apported will meet your expectations.

3. Point-by-point response to Comments and Suggestions for Authors

1.     ABSTRACT

Comments 1: The authors' intended IPC strategy for preventing outbreaks of MDR bacteria and fungi in the ICU is not clearly stated.

Response 1: Thank you for pointing this out. Due to the limited number of words given for Abstract, we could not deepen the concept further. However, IPC strategies we focused on are stated in the lines 20-22 (“We performed a narrative review on IPC in ICUs, investigating patient-to-staff ratios, education, isolation, decolonization, screening and hygiene practices, outbreak reporting, cost-effectiveness, reproduction-number (R0), and future perspectives”).

Comments 2: I suggest that the authors include their main findings in the abstract instead of making suggestions on multiple topics.

Response 2: Thank for your suggestion. Given the limited number of lines given in Abstract, we had to take a decision on how to display our findings and conclusions. As we faced different issues in the big field of IPC, and because the review nature, we decided to provide a mix of findings and conclusions drawn by the findings.

Findings:

1.     IPC strategies: no universal evidence of what the most effective IPC strategy is was found  The most effective IPC strategy remains unknown.

2.     Studies on IPC strategies: we found, when conducting this review, that the main problem when approaching this topic for physicians is the dispersion of information, as most studies focus on a specific pathogen or disease, making the clinician losing the big picture. This could contribute to the lack of IPC knowledge and consequent application by the first-line healthcare workers. As consequence, outbreak spreads.  Most studies focus on a specific pathogen or disease, making the clinician losing the big picture.

3.     Cost-effectiveness: we found out IPC strategies have resulted to be cost-effective regardless typology, country, or pathogen.  IPC strategies proved their cost-effectiveness regardless typology, country, or pathogen.

4.     HCW education: we found a large heterogeneity of IPC educational models (EM). Then, we opted for expressing in Abstract the need for a universal protocol, instead of displaying what they had in common, for example. In the text, readers can find what elements are important for an EM, together with some EM examples found in literature. A similar speech is valid for an outbreak recognition tool  A standardized, universal, pragmatic protocol for HCW education should be elaborated. A rapid outbreak recognition tool elaboration (i.e., an easy-to-use mathematical model) would improve early diagnosis and spreading prevention.

5.     MDROs decolonization: no consensus was found clearly in favor or against decolonization, with some exceptions. Therefore, more studies are needed to assess it.  Further studies are needed to express in favor or against MDROs decolonization.

6.     New strategies: we highlighted several new strategies, but due to the limited possibility to express the findings, we just invite the reader to go to apposite paragraph to find it out. And, in the last line, we mention some of them  New promising strategies are emerging and need to be tested in the field. […]. In a not-too-distant future genetic engineering and phage therapies could represent a plot-twist in MDROs IPC strategies.

Comments 3: Of all the strategies examined in the literature, which one/s is/are the most effective?

Response 3: As reported in lines 22-23, the most effective strategy remains unknown.

2.     INTRODUCTION

Comments 1: “I recommend that the authors provide additional contextual details into this particular area. What is the current status of MDROs particularly in ICU? The statistics on MDRO's and Candida auris, and their impacts in the ICU would be more useful in this context”.

Response 1: Thank you for your valuable advice. We decided to implement the text as following (lines 51-57): “In particular, MDROs’ prevalence in ICUs is a major concern nowadays as they can account up to 50% of all infections in this setting. In USA, more than 700,000 HAI each year are associated with MDR bacteria in ICU, while in Europe, a growing rate of HAI caused by Carbapenemase-producing Enterobacteriaceae (CPE) and New Delhi Metal-lo-Betalactamase (NDM) producers have been observed. However, the greatest threaten in ICU is currently represented by MDR Acinetobacter baumannii, Pseudomonas, Entero-bacteriaceae, and Candida auris, against which few are the weapons available”.

Bibliography: https://doi.org/10.1038/s41598-023-42522-2; doi: 10.3390/tropicalmed7110365

Comments 2: “The authors could discuss more on the national or international guidelines of IPC strategies and their relevancy in the ICU”.

Response 2: Thank you for the suggestion. In order not be redundant, we decided to discuss most relevant IPC guidelines in each paragraph according to the examined variable, so that we can provide more than one point of view on each variable.

Comments 3: “Why do the authors choose Candida auris in addition to MDR bacteria? It should be clear. In addition, I recommend the authors mention different types of MDR pathogens reported in the ICU”.

Response 3: Thank you for your suggestion. We mentioned Candida auris as well as the other MDR as ICU major threaten in the new lines we have added (see Comment n.1). All most frequent MDR and difficult-to-treat pathogens isolated in ICU are listed in Table 1.

Comments 4: “I suggest authors discuss the chronological scenario regarding the outbreak of MDROs, particularly in the ICU (if the data is available only). Since the authors have chosen a publication of the last 15 years, it is relevant to discuss what the situation was like 15 years ago and what it is now. How the different strategies help to reduce the outbreak and so forth.

Response 4: Thank you for the precious insight. We have provided this information in the lines added (see Comment n.1). We have chosen the 25 years cut-off in order to include only the most consolidated strategies and the novelties in the field of IPC, including those specifically against MDROs (added lines 83-85: The choice of 25 years is dictated by the will to include only the most consolidated strategies and the novelties in the field of IPC, including those specifically against MDROs). We prefer not to make a comparison between how it was before as the context was different both from an epidemiological and technological point of view. As we projected this review to the future of IPC, we preferred talking about the presence and the future of IPC. Considering “. How the different strategies help to reduce the outbreak”, we have preferred to extend these concepts in the single paragraphs, instead of in Introduction.

3.     METHODOLOGY

Comments 1: “What were the search strategies, selection criteria, and timeline of published literature?”

Response 1: Thank you for the question.

1.     Search strategies: “The search was conducted on PubMed electronic database and included only peer-reviewed articles. No language restriction was applied. Publications were firstly screened by title, abstract and year of publishing by CF. Afterwards, CF evaluated the full articles in order to assess the eligibility for inclusion, and consequently reviewed by DP, LP and PT”. (lines 79-83).

2.     Selection criteria: “The quality of data and accuracy of description of the proposed strategy, together with the novelty, were considered as the most-weighting factors in selection process”. (lines 83-84).

The type of literature examined to perform this review, include “guidelines, WHO recommendations, international institutional statements, outbreaks report in the last 25 years, although not being comprehensive of all literature as a sys-temic review would do.” (lines 74-76)

3.     Timeline of published literature: We decided to include publications dating back no more than the last 25 years (lines 75-76) in order to include only the most consolidated strategies and the novelties in the field of IPC. In order to provide an explanation for the readers as well, we decided to implement the Methodology with this few lines: “The choice of 25 years is dictated by the will to include only the most consolidated strategies and the novelties in the field of IPC, including those specifically against MDROs”

Comments 2: “What were the inclusion and exclusion criteria of the available literature?”

Response 2: Thank you for the question. We included only peer-review literature (line 80), comprehensive of “guidelines, WHO recommendations, international institutional statements, outbreaks report in the last 25 years” (lines 74-75). The quality of data and accuracy of description of the proposed strategy, together with the novelty, were considered as the most-weighting factors in selection process (inclusion/exclusion) operated by CF and revised by DP, LP, and PT. 

Comments 3: “How were the quality assessment and risk of bias determined?”

Response 3: Thank you for the question. The quality of data and accuracy of description of the proposed strategy, together with the novelty, were considered as the most-weighting factors in selection process (inclusion/exclusion) operated by CF and revised by DP, LP, and PT. In order to assure the good quality of literature selection and address for possible bias, only peer-reviewed articles (line 80) were included. As the “narrative approach” is non-systematic, there are no acknowledged formal guidelines for writing narrative reviews (https://doi.org/10.1016/j.hlc.2018.03.027).

4.     STRATEGIES

Comments 1: “How did the authors come upon the variables listed as important to consider throughout the infection control process?”

Response 1: Thank you for your valuable question. It is my pleasure to show you how variables were selected on multiple basis:

·       Most IPC problematic issues faced daily in real-life in CF experience as Infectious Diseases specialist and ICU consultant, and DP, LP, and PT experience as ICU specialists;

·       Most recurrent IPC strategies mentioned in IPC guidelines and WHO statements;

·       Most frequently discussed strategies, approaches and solutions reported in outbreak reports.

In order to cope with the question you asked, we decided to add the following sentence in lines 85-90 in Methodology section: “All investigated variables in the context of IPC (patient-to-staff ratios, education, isolation, decolonization, screening and hygiene practices, outbreak reporting, cost-effectiveness, reproduction-number(R0), and future perspectives) were selected on the basis of 1) recurrency in IPC guidelines and WHO statements, 2) frequency of discussion and solution proposals in outbreak reports, together with 3) real-life, daily-faced IPC problematic issues authors experience as infectologist (CF) and intensivists (DP, LP, PT).”

Comments 2: “How do the authors know they haven't overlooked any important factors?”

Response 2: We agree with your comment. Of course, we cannot be sure not to have overlooked any important factor. In order to underline this weakness of the narrative nature of our review, we will implement limitations with the following sentence (line 751-752):

“This implies that we might have unconsciously overlooked other variables impacting on IPC in our review”.

5.     OTHER SUGGESTIONS

Comments 1: “The manuscript must be proofread for every tiny error. As an illustration, consider writing Candida auris in italics, pr the complete form of all acronyms, and so on”.

Response 1: Thank you for detecting the typos. We have looked for them and fixed them.

Comments 2: “Remove all abbreviations from the heading”.

Response 2: We agree with your suggestion. We removed the abbreviations from main headings (strategies) with the exception of the abbreviation for MDRO, that we decided to write extensively in the first line after the headings, so that the meaning is immediately clear for the reader.

Comments 3: “English language and style are fine, but a throughout check is required”.

Response 3: Thank you for the suggestion. I am in possess of a Certificate in Advanced English (C1) released by the University of Cambridge and I have performed the Certificate of Proficiency in English (C2) test, but, in consideration to your comment, I have submitted the paper to an English mother tongue as to verify the appropriateness of the English language.

4. Additional clarifications

We care about the quality of the content as well as the quality of how the content is displayed, so we particularly appreciated your comments.

Agree. I/We have, accordingly, done/revised/changed/modified…..to emphasize this point. Discuss the changes made, providing the necessary explanation/clarification. Mention exactly where in the revised manuscript this change can be found – page number, paragraph, and line.]

“[updated text in the manuscript if necessary]”

Round 2

Reviewer 2 Report

Comments and Suggestions for Authors

The authors acknowledged the comments well.